**Data Availability Statement:** The minimal anonymised dataset is available from Mendeley (DOI: 10.17632/whbk485wbw.1). The data

# Cohort profile: The Chikwawa lung health cohort; a population-based observational non-communicable respiratory disease study of adults in Malawi

**Martin W. Njoroge**[1,2]*, **Sarah Rylance**[1,2], **Rebecca Nightingale**[1,2], **Stephen Gordon**[1,2], **Kevin Mortimer**[1], **Peter Burney**[3], **Jamie Rylance**[1,2], **Angela Obasi**[1], **Louis Niessen**[1,4], **Graham Devereux**[1], on behalf of The IMPALA Consortium[¶]

1 Liverpool School of Tropical Medicine, Liverpool, United Kingdom, 2 Malawi Liverpool Wellcome Trust Programme, Blantyre, Malawi, 3 National heart and Lung Institute, Imperial College, London, United Kingdom, 4 John Hopkins Bloomberg School of Public Health, Baltimore, MD, United States of America

¶ Membership of the IMPALA Consortium is provided in the Acknowledgments.
* martin.njoroge@lstmed.ac.uk

## Abstract

### Purpose

The aim of this article is to provide a detailed description of the Chikwawa lung health cohort which was established in rural Malawi to prospectively determine the prevalence and causes of lung disease amongst the general population of adults living in a low-income rural setting in Sub-Saharan Africa.

### Participants

A total of 1481 participants were randomly identified and recruited in 2014 for the baseline study. We collected data on demographic, socio-economic status, respiratory symptoms and potentially relevant exposures such as smoking, household fuels, environmental exposures, occupational history/exposures, dietary intake, healthcare utilization, cost (medication, outpatient visits and inpatient admissions) and productivity losses. Spirometry was performed to assess lung function. At baseline, 56.9% of the participants were female, mean age was 43.8 (SD:17.8) and mean body mass index (BMI) was 21.6 Kg/m$^2$ (SD: 3.46)

### Findings to date

The cohort has reported the prevalence of chronic respiratory symptoms (13.6%, 95% confidence interval [CI], 11.9–15.4), spirometric obstruction (8.7%, 95% CI, 7.0–10.7), and spirometric restriction (34.8%, 95% CI, 31.7–38.0). Additionally, an annual decline in forced expiratory volume in one second [FEV$_1$] of 30.9mL/year (95% CI: 21.6 to 40.1) and forced vital capacity [FVC] by 38.3 mL/year (95% CI: 28.5 to 48.1) has been reported.

collection tools are provided as Supporting Information files. Further information about the data can be obtained from the corresponding author (martin.njoroge@lstmed.ac.uk). All the data from the Chikwawa lung health cohort presented in this article are stored by the research group on safe servers at the Malawi Liverpool Wellcome Trust programme (MLW), Malawi and the BOLD centre at Imperial College London, UK and handled confidentially.

**Funding:** This research was funded by the National Institute for Health Research (NIHR) (project reference 16/136/35) using UK aid from the UK Government to support global health research. The views expressed in this publication are those of the author(s) and not necessarily those of the NIHR or the UK Department of Health and Social Care. Initial funding (between 2013 – 2017) was provided by a New Investigator Research Grant from the Medical Research Council (Ref: MR/L002515/1), a Joint Global Health Trials Grant from the Medical Research Council, UK Department for International Development and Wellcome Trust (Ref: MR/K006533/1).

**Competing interests:** The authors have declared no competing interests.

## Future plans

The ongoing phases of follow-up will determine the annual rate of decline in lung function as measured through spirometry and the development of airflow obstruction and restriction, and relate these to morbidity, mortality and economic cost of airflow obstruction and restriction. Population-based mathematical models will be developed driven by the empirical data from the cohort and national population data for Malawi to assess the effects of interventions and programmes to address the lung burden in Malawi. The present follow-up study started in 2019.

## Introduction

Globally, non-communicable respiratory diseases (NCRD) are the third leading cause of non-communicable disease (NCD) mortality, causing an estimated 4 million deaths each year [1]. Amongst the NCRD, asthma and chronic obstructive pulmonary disease (COPD) are the most prevalent, affecting approximately 358 million and 174 million people respectively [2]. Annually, COPD causes 3 million deaths accounting for 6% of all deaths worldwide [2–4]. Furthermore, the deaths from these diseases are rising globally [5] in part due to increased longevity and changes in population structure [6].

The majority of the burden of NCRD mortality and morbidity is in low and middle-income countries (LMIC) [1,7], which now account for 90% of COPD deaths [8]. Several community based studies in LMIC have documented a high prevalence of abnormal lung function, both obstructive and restrictive (low lung volumes) [9–15], whilst several couple have documented low prevalence of COPD [16,17] but high prevalence of respiratory symptoms [17]. In contrast, very few observational cohort studies have reported and described the health and economic burden of NCRD [18,19], especially in LMIC settings. Their prevalence means that there is a pressing need to better document the life course epidemiology and the related health and economic burden of abnormal (obstructive and restrictive) lung function in LMIC [10,11].

Malawi remains one of the poorest countries in the world [20] with 83% of its 18 million inhabitants living in rural areas [21]. With a GDP per capita of $300, over half the households live below the poverty line (using the international poverty line of US$ 1.90 per person per day) [22], and about 50% of the national health expenditure is funded from external donors [23,24]. In common many sub-Saharan African (SSA) countries, Malawi is at the intersection of high rates of communicable respiratory diseases (Tuberculosis (TB), pneumonia), and increasing NCRD [25–27]. Although Malawi has a well-established TB control programme, only 10–20% of patients presenting at primary healthcare facilities with a persistent cough have TB [28]. The prevalence of diagnosed NCRD such as COPD, asthma and pulmonary fibrosis is essentially unknown [29] because lung function testing is not available out with research settings [30]. In Chikwawa where this study is based, lung function testing is not available at primary health facilities or secondary care (Chikwawa District Hospital). There is very limited capacity to perform spirometry in tertiary care, and this is provided by research staff, and for most patients, transport cost would prevent them from travelling to access this.

Recently, however, studies have reported substantial levels of abnormal lung function in Malawi, with spirometric evidence of restrictive and obstructive deficits present in 34.8% (95% CI: 31.7%, 38.0%) and 8.7% (7.0%, 10.7%) of rural adults and 38.6% (34.4%, 42.8%) and 4.2% (2.0%, 6.4%) of urban adults respectively [10,11]. Spirometric deficits were defined according

to the NHANES III Caucasian references [31]. What is not known is, whether, and how these spirometric deficits impact on the everyday lives of the country's people and health system. Potentially, as in other low-income situations, the economic burden of NCRD may have serious adverse outcomes for households including unpredictable household expenditures due to complications and catastrophic health expenditure [32].

To examine the health and economic burden of NCRD, including abnormal lung function in Malawi, our prospective study aims to follow up a population-based cohort of participants in the rural district of Chikwawa, in southern Malawi, who were recruited to a longitudinal follow-up spirometry study conducted between August 2014 and July 2015 (the Chikwawa lung health cohort) [11,15]. The primary objectives of the current study are to; (i) estimate the annualised rate of change in lung function by age and sex as determined by repeating spirometry; (ii) to develop a mathematical population model based on the cohort findings that estimates the life-time health impact of airflow obstruction in Malawian adults in disability-adjusted life years (DALYS); (iii) estimate the health resource use and lifetime costs in the cohort of Malawian adults with airflow obstruction in international dollars (Int$); (iv) produce model estimates of the lifetime cost effectiveness (Int$/DALY) of selected key intervention compared with current practice to define optimum packages of interventions; and (v), recreate these analyses for Malawian adults with low lung volumes. The economic cost will be from a societal perspective and will include health sector costs, patient/family and carer costs and productivity losses [33]. Presently, the Malawian health system recommends the use of salbutamol and beclomethasone inhalers and prednisolone as interventions for chronic asthma management and salbutamol inhalers, prednisolone and hydrocortisone injections as interventions for acute asthma [34] but these interventions are only available in 8% of urban health facilities and 2% of rural health facilities in Malawi [35]. The current study will determine whether the substantial levels of abnormal lung function in Malawian adults are clinically and societally important not only currently, but also in the future by estimating the economic cost of obstructive and restrictive conditions. In addition, the present study will provide a basis for NCRD intervention adoption and implementation within the Malawian health system, society and similar settings.

The aim of this cohort profile paper is to provide a comprehensive description of the Chikwawa lung health cohort as a research resource for potential collaborations, including an overview of the collected data, a description of the baseline characteristics and a summary of the main results published so far.

## Cohort description and methods

### Setting

The study is currently conducted in Chikwawa district, located in Southern region of Malawi. (see Fig 1).

### Study population

The Chikwawa lung health cohort was initiated alongside the Cooking and Pneumonia study (CAPS) [11,36] (Trial registered with ISRCTN, number ISRCTN59448623). CAPS was a cluster randomized trial that investigated the health effects of a cleaner-biomass fuel cookstove intervention [36]. The aim of setting up the Chikwawa lung health cohort was to determine the prevalence and determinates of lung disease amongst adults in Chikwawa, rural Malawi [11]. Since inception, two rounds of follow-up studies have been conducted with the Chikwawa lung health cohort aiming to assess the determinants of lung function trajectories as affected by personal air pollutant exposures, including the CAPS cookstove intervention [15].

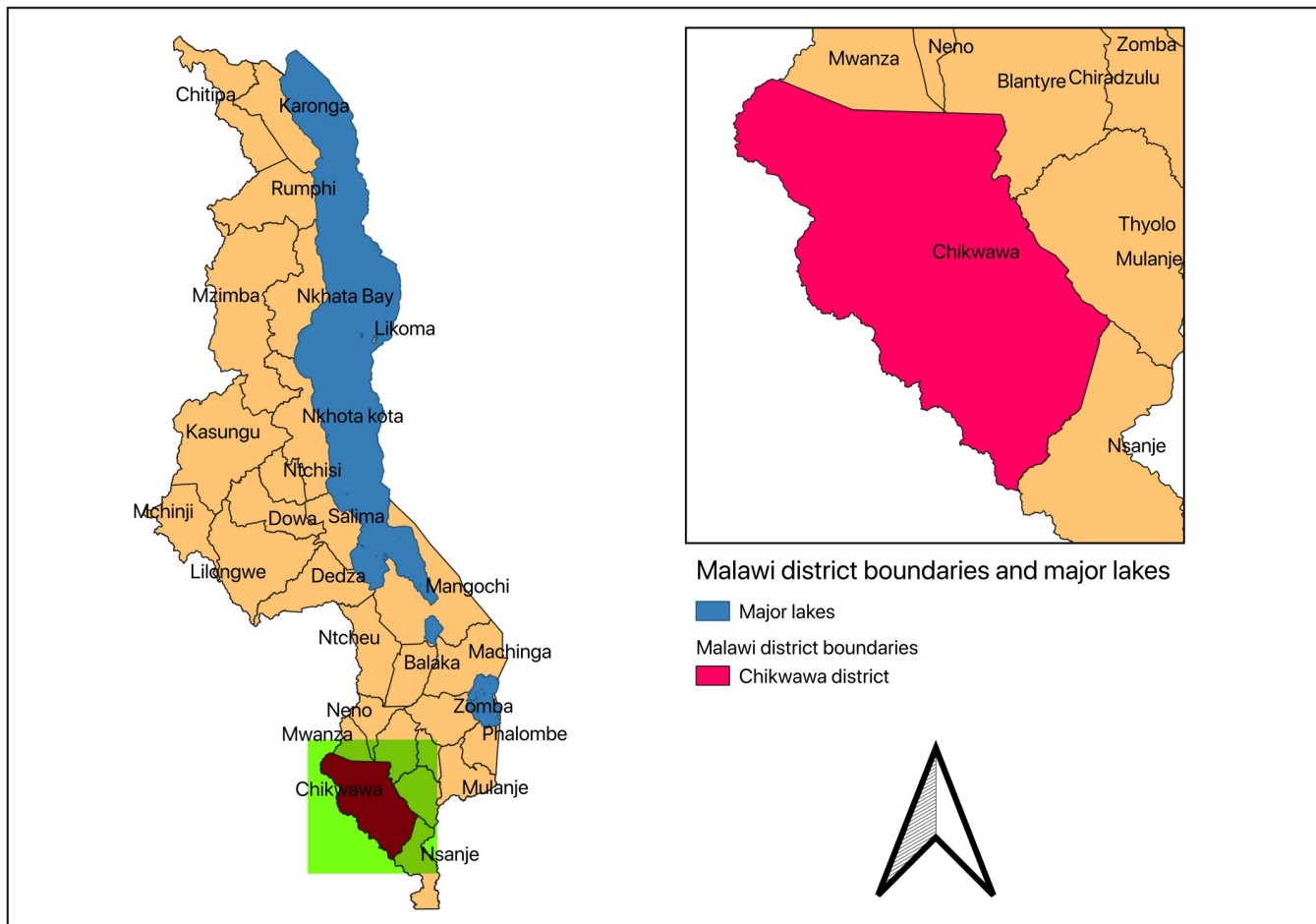

**Fig 1. Districts in Malawi.** Inset map highlights Chikwawa district, the study area. (Created using the open source QGIS ver. 3.8. Zanzibar (QGIS Development Team, 2020, https://qgis.org.

The current study will provide additional longitudinal data by further following up participants from the Chikwawa lung health cohort who still reside in Chikwawa and who were recruited to the baseline study in 2014–2015 [11] and quantify associated risk factors, health utilisation use and economic burden.

## Statistical analyses

The sociodemographic and clinical variables were determined for the sample using frequencies and proportions for categorical variables and means and standard deviation for continuous variables. Chi-square and t test were used to investigate associations between gender and the other variables.

## Baseline participant recruitment

The participants were originally recruited in 2014–2015. The participants were selected through random sampling of a list of adults living in each of the 50 villages participating in CAPS [11]. The participants included those who took part in the CAPS intervention and those who did not but resided in villages where the CAPS intervention was being implemented. The list of adults was obtained from local community liaison personnel from each village following

**Table 1. Demographic characteristics of cohort participants.**

| | | Consenting participants n = 1481 | Selected, did not give consent n = 1519 |
|---|---|---|---|
| Age, mean (SD) | | 43.9 (17.8) | 40.3 (16.5) |
| Age categories years n (%) | <39 | 685 (46.3%) | 765 (50.3%) |
| | 40–49 | 258 (17.4%) | 336 (22.1%) |
| | 50–59 | 217 (14.7%) | 179 (11.8%) |
| | 60–69 | 161 (10.9%) | 150 (9.9%) |
| | >70 | 160 (10.8%) | 89 (5.9%) |
| Sex | Female | 844 (57.0%) | 757 (49.9%) |
| | Male | 637 (43.0%) | 762 (50.2%) |

a series of community engagement events with the village leaders such as chiefs and other community representatives [11]. The random selection was conducted by an independent statistician at the Burden of Obstructive Lung Disease (BOLD) centre in London in accordance with the BOLD protocol [37]. The identified individuals comprised a population-representative, age and gender stratified, sample of adults who were then invited to participate in the 2014–2015 baseline study. Participants had to provide written informed consent or an independently witnessed thumbprint to be included in the study [11]. Those who were acutely unwell or pregnant women or were non-permanent residents of Chikwawa were excluded from the baseline study [11].

A total of 3000 adults were invited to participate in the baseline study of which 1481 (49.3%) agreed to participate [11]. Participants were stratified into two age groups: 18–39 years and 40 years and above. In order to provide an estimate of chronic airflow limitation prevalence in the stratum with a precision (95% CI) of +3.3% to 5.0% and assuming a prevalence of 10% to 25%, a total sample of 1200 participants was estimated allowing for unequal age and gender distribution, refusals and inability to provide spirometry measurements of acceptable quality [11]. Table 1 below summarises the age and sex characteristics of those who agreed to participate in the study compared to those who did not.

## Participant tracking and recruitment procedures for the current longitudinal study

In the current study, the adult participants have been tracked from participant logs developed in the original baseline study [11]. The participant log contains the person's name, study identification number, age, gender and village of residence. Community liaison personnel and chiefs were asked to help identify the household of each study participant to maximise fidelity. Study staff then approached the participant in their households, obtained informed consent, geolocation, and agreed a suitable time to collect the lung function, environmental exposures and socioeconomic data.

## How often has the cohort been followed up?

Study participants have been followed up twice prior to the current study. The baseline study was conducted between August 2014 –July 2015 [11] with an aim of determining the prevalence and determinates of lung disease. The first and second follow-up studies were between August 2015 –November 2017 [15] aiming to assess the determinants of lung function trajectories as affected by personal air pollutant exposures, including the CAPS cookstove intervention. The current round of follow-up is taking place between July 2019 –March 2021 (see Fig 2).

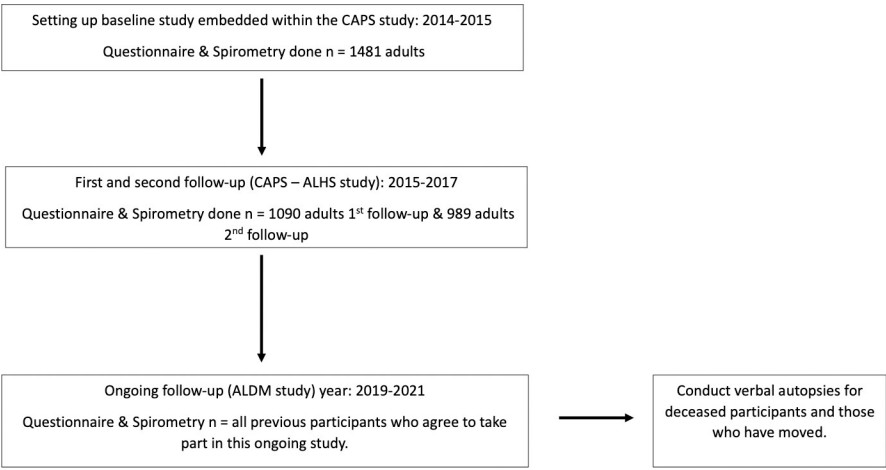

**Fig 2. Flow chart of participant recruitment and follow-up schedule.**

## Assessment of exposures

In the baseline study, structured interviews were used to collect data on demographic, socio-economic status, respiratory symptoms, and potentially relevant exposures such as smoking [38,39], household fuels [38,40], environmental exposures [39,41], and occupational history [39,42].

In addition to the data collected for the baseline study in the current 2019–2021 follow up, we are collecting additional data on dietary intakes [39,43], healthcare utilization, cost (medication, outpatient visits and inpatient admissions) and productivity losses.

The following anthropometric measures have been recorded at each phase of follow up: height, weight, hip, waist, and neck circumferences, ulna, and fibula lengths. Lung function (forced expiratory volume in one second [$FEV_1$] and forced vital capacity [FVC]) are measured using the ndd EasyOne Spirometer (ndd Medizintechnik AG, Zurich, Switzerland), before and 15 minutes after administration of inhaled salbutamol (200 μg) administered via spacer device. The contraindications for spirometry include: in the previous three months; thoracic or abdominal surgery, acute coronary syndrome, detached retina or eye surgery; hospitalisation for any other cardiovascular reason in the previous month; final trimester of pregnancy; a resting heart rate > 120 beats per minute and current treatment for tuberculosis [44].

Spirometry has been conducted by trained and certified technicians who received regular feedback on spirogram quality in accordance with the BOLD protocol [37]. The quality of each spirogram has been reviewed and scored based on the American Thoracic Society and European Respiratory Society acceptability and reproducibility criteria [45].

In the current phase of follow-up, verbal autopsies were conducted for the 2014–2015 baseline participants who have died, and a questionnaire was administered to the next of kin for those who were unobtainable due to being no longer resident in Chikwawa. The data and variables collected in the Chikwawa lung health cohort are described in Table 2.

## Ethical approval

The study protocol was approved by the Imperial College Research Ethics Committee (17IC4272) (website: https://www.imperial.ac.uk/research-ethics-committee/committees/icrec/), Liverpool School of Tropical Medicine Research Ethics Committee (19–005) (website: https://www.lstmed.ac.uk/research/research-integrity/research-ethics-committee) and the

**Table 2. Summary of measurements in the Chikwawa lung health cohort.**

| Phase | Spirometry measured | Anthropometric measured | Questionnaires & tools administered |
|---|---|---|---|
| Baseline 2014–2015 [11] | • Forced vital capacity (FVC)<br>• Forced expiratory volume in 1 second ($FEV_1$)<br>• Forced expiratory volume in 6 seconds ($FEV_6$) | • Weight<br>• Height<br>• Waist & hip circumference | • Socio-economic status<br>• Demographic characteristics<br>• Environmental exposures<br>• Smoking history<br>• History of respiratory disease (Tuberculosis, Asthma and COPD). |
| First and second follow-up (this follow-up phase was called the CAPS-Adult Lung Health study) 2015–2017 [15] | • FVC<br>• $FEV_1$<br>• $FEV_6$. | • Weight<br>• Height<br>• Waist & hip circumference | • Socio-economic status<br>• Demographic characteristics<br>• Environmental exposures<br>• Smoking history<br>• History of respiratory disease (Tuberculosis, Asthma and COPD).<br>• Personal air pollutant monitoring |
| Ongoing (this follow-up phase is called Adult Lung Diseases in Malawi study) 2019–2021 | • FVC<br>• $FEV_1$<br>• $FEV_6$. | • Weight,<br>• Height;<br>• Ulna & fibula lengths;<br>• Neck, waist & hip circumference. | • Socio-economic status<br>• Demographic characteristics<br>• Environmental exposures<br>• Smoking history<br>• History of respiratory disease (Tuberculosis, Asthma and COPD)<br>• History of health utilization and costs (medication, outpatient & inpatient)<br>• Productivity losses<br>• Household dietary consumption |

Malawi College of Medicine Research and Ethics Committee (COMREC, P.03/19/2617) (website: https://www.medcol.mw/college-of-medicine-research-ethics-committee/). Written informed consent was obtained from all the participants in this study for the follow-up and for the second interview and examination.

## Participant and public involvement

Participants were not involved in in setting research questions or the outcome measures but have been instrumental in implementation of the study.

Participants and the public were involved in the dissemination of baseline information nationally through the Ministry of Health, and in the Chikwawa community from which the data was collected through the Chikwakwa Health Research Committee and the Chiefs and community leaders from the villages from where we collected our data. These activities have encouraged community buy-in and involvement in the subsequent rounds of follow-up within the cohort.

## Findings to date and discussions

The Chikwawa lung health cohort has provided data characterising the burden of chronic respiratory symptoms, abnormal spirometry and air pollution exposures and risk factors from an adult population in Malawi [11,36]. These data have contributed to the understanding of NCRD in LMIC. The baseline characteristics of the Chikwawa lung health cohort when established in 2014–2015 are outlined in Table 3. At baseline, a total of 1481 participants were recruited of which 637 (43.0%) were male and 844 (57.0%) were female [11]. The mean age was 43.9 years (SD: 17.8), mean body mass index (BMI) was 21.6 Kg/m$^2$ (SD: 3.46). Cigarette smoking rates were 22.1% (n = 327) were current or ever smokers of which the majority were men (n = 255, 78.0%). There was no difference in ages between the men and women (see Table 3).

**Table 3. Baseline demographic, anthropometric and symptomatic characterises of the Chikwawa lung health cohort collected 2014–2015.**

| Variable (n) | | n (%) (total = 1481) | Male n (%) (total = 637) | Female n (%) (total = 844) | P value (X²)[Ψ] |
|---|---|---|---|---|---|
| Age group (years) | <39 | 685 (46.3%) | 288 (45.2%) | 397 (47.0%) | 0.150 |
| | 40–49 | 258 (17.4%) | 103 (16.2%) | 162 (18.4%) | |
| | 50–59 | 217 (14.7%) | 110 (17.3%) | 110 (12.7%) | |
| | 60–69 | 161 (10.9%) | 70 (11.0%) | 96 (10.8%) | |
| | >70 | 160 (10.8%) | 66 (10.4%) | 99 (11.1%) | |
| BMI [##] | Underweight (< 18.5) | 182 (13.9%) | 84 (13.2%) | 98 (11.6%) | <0.001 |
| | Normal weight (≥18.5; <25.0) | 950 (72.8%) | 465 (73.0%) | 485 (57.5%) | |
| | Overweight (≥25.0; <30.0) | 133 (10.2%) | 36 (5.7%) | 97 (11.5%) | |
| | Obese (≥ 30.0) | 40 (3.1%) | 2 (0.3%) | 38 (4.5%) | |
| Smoking | Never | 1154 (77.9%) | 382 (60.0%) | 772 (91.5%) | <0.001 |
| | Current | 205 (13.8%) | 165 (25.9%) | 40 (4.7%) | |
| | Former | 122 (8.2%) | 90 (14.1%) | 32 (3.8%) | |
| **Symptoms** | | | | | |
| Cough on most days of the month for at least three months of the year. | | 167 (11.3%) | 81 (12.7%) | 86 (10.2%) | 0.148 |
| Usually brings up phlegm from chest | | 39 (2.6%) | 21 (3.3%) | 18 (2.1%) | 0.221 |
| Wheezing/whistling in chest in the past 12 months in the absence of a cold. | | 24 (1.6%) | 15 (2.4%) | 9 (1.1%) | 0.082 |
| MRC dyspnoea score II [46,47]: shortness of breath when hurrying on the level or walking up a slight hill. | | 23 (1.6%) | 11 (1.7%) | 12 (1.4%) | 0.766 |
| Any respiratory symptoms | | 203 (13.7%) | 105 (16.5%) | 98 (11.6%) | 0.008 |
| Functional limitation: breathing problems interfere with usual daily activities. | | 44 (3.0%) | 21 (3.3%) | 23 (2.7%) | 0.624 |
| **Diagnosed lung disease** | | | | | |
| Asthma | | 51 (3.4%) | 23(3.6%) | 28 (3.3%) | 0.868 |
| Asthma, emphysema, chronic bronchitis, or COPD | | 59 (4.0%) | 28 (4.4%) | 31 (3.7%) | 0.566 |
| Previous TB | | 47 (3.2%) | 16 (2.5%) | 31 (3.7%) | 0.268 |

[##] n = 1341. BMI classification based on WHO guidelines [48].

[Ψ] Comparison of proportions using Pearson's chi square test.

## The frequency of chronic respiratory symptoms and abnormal spirometry

Among the participants at recruitment in 2014–15, with interpretable and reliable spirometry (n = 886) [37], spirometric obstruction (defined as $FEV_1/FVC < 0.70$) and spirometric restriction (defined as $FEV_1/FVC > 0.70$ and post-bronchodilator FVC < 80% predicted) [31] were present in 8.7% (95% CI: 7.0%, 10.7%) and 34.8% (95% CI: 31.7%, 38.0%) of the participants respectively according to the NHANES III Caucasian references [11]. 13.7% reported either having a 'cough without having a cold', 'bringing up phlegm from your chest', 'wheezing in your chest', 'shortness of breath when hurrying on the level or walking up a slight hill', or 'breathing problems interfering with your daily activity' while 11.3% reported a 'cough on most days of the month for at least three months per year'. 3.4% were diagnosed with asthma while 4.0% were diagnosed with either asthma, emphysema, chronic bronchitis, or COPD (see Table 3). The 2017 follow-up found that, when compared to the NHANES III African American reference ranges, spirometric obstruction and restriction were present in 11.5% (95% CI: 9.6%, 13.5%) and 7.7% (95% CI: 6.2%, 9.5%) of the participants respectively [15]. For participants who had been followed up in both 2015 to 2017, an overall annual rate of lung function decline in forced expiratory volume in one second [$FEV_1$] of 30.9mL/year (95% CI: 21.6 to

40.1) and forced vital capacity [FVC] by 38.3 mL/year (95% CI: 28.5 to 48.1) has been reported [15].

Presently, we are able to trace over 85% of the participants in the Chikwawa lung health cohort and have invited them to participate in this current phase of follow-up. The ongoing analysis of the data at a later time point for follow up will provide better estimates for annual rate of lung function decline.

### Present research plans

The ongoing current phase of follow-up of the Chikwawa lung health cohort will determine the annual rate of decline in lung function as measured through spirometry, morbidity, mortality and economic cost of airflow obstruction and restriction and develop population-based mathematical models driven by the empirical data from the cohort and national population data for Malawi to assess the effects of interventions and programmes to address the lung burden in Malawi. It is expected that this further phase of follow-up will add to the body of knowledge of the life course of NCRD in LMIC and further refine and add to the validity of the health economic models developed.

### Strengths and limitations

The Chikwawa lung health cohort appears to be the only one of its kind in a low-income country setting aiming to investigate the economic costs over the life course of non-communicable respiratory disease. This cohort represents an opportunity to develop and model cost-effective interventions and programmes for this setting. The baseline cohort was conducted alongside a rigorously conducted cluster randomised control trial. Despite local complexities, we presently have identified over 85% of the baseline cohort to be included in the current phase of follow-up.

Systematic bias may be introduced through the self-selection of the participants who agreed to take part in the study to date and the migration of individuals from Chikwawa. Although we have been able to track over 85% of the original Chikwawa lung health cohort and have invited them to participate in the current phase of follow-up, the participants who can be traced and from whom data are collected may differ from those who cannot be traced or do not attend follow-up. Similarly, at baseline, the participants who agreed to be consented were slightly older and mainly women. The process of verbal autopsies for those who have died [49], and collection of data from the next of kin of those who have moved away, may shed some light on the status of those who have moved away from Chikwawa and deaths from respiratory causes will be of particular interest in the current follow-up. The other limitation identified in this study is recall bias. This is due to most of the data being collected through administering questionnaires in a structured interview format, one can expect recall bias over the follow-up period. We are using tested and validated tools in addition to well-trained experienced interviewers to minimize this bias.

The main strength of the cohort is the collection of initial objective measures of lung function using spirometry conducted to internationally agreed standards [37,45] and on two further occasions over a 3-year period. This will provide valuable insights into the health relevance and natural history of abnormal lung functions in an LMIC setting. Previous studies in the United States, the United Kingdom and Australia have reported the annual rate of decline of $FEV_1$ in adults to be 18 ml/year standard deviation (SD) = 2.5 [50], 33ml/year (SD = 1.5) [51] and 45ml/year (SD = 83) [52].

## Supporting information

**S1 File. Link to minimal anonymized dataset.**
(DOCX)

**S2 File.**
(PDF)

## Acknowledgments

We would like to thank all the people involved in the collection of data for the Chikwawa lung health cohort and all the cohort participants. We would like to thank Patrick Mjojo, Catherine Chirwa, Frank Jonas and Chifundo Mhango for their immense assistance during data collection in this present round of follow-up. MN, KM, JR, AO, LN and GD are members of the IMPALA consortium.

## Author Contributions

**Conceptualization:** Martin W. Njoroge.

**Data curation:** Martin W. Njoroge, Sarah Rylance, Rebecca Nightingale.

**Formal analysis:** Martin W. Njoroge.

**Funding acquisition:** Kevin Mortimer, Graham Devereux.

**Project administration:** Martin W. Njoroge.

**Resources:** Peter Burney, Jamie Rylance.

**Supervision:** Stephen Gordon, Kevin Mortimer, Jamie Rylance, Angela Obasi, Louis Niessen, Graham Devereux.

**Validation:** Stephen Gordon, Kevin Mortimer, Peter Burney, Graham Devereux.

**Writing – original draft:** Martin W. Njoroge.

**Writing – review & editing:** Martin W. Njoroge, Sarah Rylance, Rebecca Nightingale, Stephen Gordon, Kevin Mortimer, Peter Burney, Jamie Rylance, Angela Obasi, Louis Niessen, Graham Devereux.

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
