## [Decision Letter · Decision Letter 0]

21 Sep 2020

PONE-D-20-12808

Cohort profile: The Chikwawa lung health cohort; a population-based observational non-communicable respiratory disease study of adults in Malawi.

PLOS ONE

Dear Dr. Njoroge,

Thank you for submitting your manuscript to PLOS ONE. After careful consideration, we feel that it has merit but does not fully meet PLOS ONE’s publication criteria as it currently stands. Therefore, we invite you to submit a revised version of the manuscript that addresses the points raised during the review process.

We look forward to receiving your revised manuscript.

Kind regards,

Alana T Brennan

Academic Editor

PLOS ONE

Journal Requirements:

2. Please include additional information regarding the interview guide and questionnaire used in the study and ensure that you have provided sufficient details that others could replicate the analyses. For instance, if you developed a questionnaire as part of this study and it is not under a copyright more restrictive than CC-BY, please include a copy, in both the original language and English, as Supporting Information.

3.We note that [Figure(s) 1] in your submission contain [map/satellite] images which may be copyrighted. All PLOS content is published under the Creative Commons Attribution License (CC BY 4.0), which means that the manuscript, images, and Supporting Information files will be freely available online, and any third party is permitted to access, download, copy, distribute, and use these materials in any way, even commercially, with proper attribution. For these reasons, we cannot publish previously copyrighted maps or satellite images created using proprietary data, such as Google software (Google Maps, Street View, and Earth). For more information, see our copyright guidelines: http://journals.plos.org/plosone/s/licenses-and-copyright.

1.    You may seek permission from the original copyright holder of Figure(s) [1] to publish the content specifically under the CC BY 4.0 license. 

4.We note that you have indicated that data from this study are available upon request. PLOS only allows data to be available upon request if there are legal or ethical restrictions on sharing data publicly. For information on unacceptable data access restrictions, please see http://journals.plos.org/plosone/s/data-availability#loc-unacceptable-data-access-restrictions.

Reviewers' comments:

Reviewer's Responses to Questions

**Comments to the Author**

1. Is the manuscript technically sound, and do the data support the conclusions?

Reviewer #1: Yes

Reviewer #2: Yes

2. Has the statistical analysis been performed appropriately and rigorously? 

Reviewer #1: No

Reviewer #2: Yes

3. Have the authors made all data underlying the findings in their manuscript fully available?

Reviewer #1: No

Reviewer #2: No

4. Is the manuscript presented in an intelligible fashion and written in standard English?

Reviewer #1: Yes

Reviewer #2: Yes

5. Review Comments to the Author

Reviewer #1: 1. An aim of the baseline study was to determining the prevalence and determinates of lung disease. Please clarify the prevalence of major lung diseases in this study, especially those that cause spirometric restriction.

2. For such a cohort study, it might make more sense to add the following: What is the diagnostic rate of COPD and asthma? How aware is the population of these diseases? How popular are lung function tests?

3. Analysis of lung function decline in different diseases and age stratification.

4. Analysis of reasons for reluctance to participate in follow-up.

5. There are obvious errors in the statistics of many demographic data, which need to be further verified in Table 3.

Reviewer #2: Purpose well stated but no justification- include justification (The WHY)

Please include names of Ethical review members

What was the involvement of local researcher

NHANES III – used no mention of limitation/ if appropriate or how it was adjusted for non-Caucasian population.

With over 50% of participants decline to be recruited – there should be information of why? And state if this will bias the study

please include results of lung function test done

Limitation

Recall bias -using tested and validated tools is a strong point.

‘ 13.7% reported either having a ‘cough without having a cold’, ‘bringing up phlegm from your chest’, ‘wheezing in your chest’, ‘shortness of breath when hurrying on the level or walking up a slight hill’, or ‘breathing problems interfering with your daily activity’ ‘- not clear where the figure 13.7% was derived, (was it and or?) please clarify

6. PLOS authors have the option to publish the peer review history of their article (what does this mean?). If published, this will include your full peer review and any attached files.

Reviewer #1: No

Reviewer #2: No

---

## [Author Response · Author response to Decision Letter 0]

12 Oct 2020

12th October 2020. 

The Editorial office,

PLOS One.

Dear Editors,

RE: Cohort Profile: The Chikwawa lung health cohort; a population-based observational respiratory disease study in Malawi.

Thank you for your review of the above titled manuscript. This is a response to the queries that were raised through the email received on the 21st September 2020.

The responses are separated by journal and reviewer comments and highlighted in red:

This has been done; however, we would be more than happy to make further changes if deemed necessary by the editorial team.

2) Please include additional information regarding the interview guide and questionnaire used in the study and ensure that you have provided sufficient details that others could replicate the analyses. For instance, if you developed a questionnaire as part of this study and it is not under a copyright more restrictive than CC-BY, please include a copy, in both the original language and English, as Supporting Information

The interview questionnaires have been added as supplementary files to the manuscript and labelled accordingly.

3) We note that [Figure(s) 1] in your submission contain [map/satellite] images which may be copyrighted. All PLOS content is published under the Creative Commons Attribution License (CC BY 4.0), which means that the manuscript, images, and Supporting Information files will be freely available online, and any third party is permitted to access, download, copy, distribute, and use these materials in any way, even commercially, with proper attribution. For these reasons, we cannot publish previously copyrighted maps or satellite images created using proprietary data, such as Google software (Google Maps, Street View, and Earth).

This map was created using the open source QGIS ver. 3.8. Zanzibar (QGIS Development Team, 2020, https://qgis.org). The authors’ own the copyright and PLOS can publish the figure under the Creative Commons Attribution License (CC BY 4.0).

4) We note that you have indicated that data from this study are available upon request. PLOS only allows data to be available upon request if there are legal or ethical restrictions on sharing data publicly.

During the review process at PLOS One, a minimal anonymized dataset for this study was published on Mendeley and is publicly accessible. The DOI has been included in the publication in the supplementary information (S1 File). The data collection tools have been provided in the supplementary information (S2 File).

5) Your ethics statement should only appear in the Methods section of your manuscript. If your ethics statement is written in any section besides the Methods, please move it to the Methods section and delete it from any other section. Please ensure that your ethics statement is included in your manuscript, as the ethics statement entered into the online submission form will not be published alongside your manuscript.

This has been done.

Reviewers’ comments

Reviewer #1: 

1) An aim of the baseline study was to determining the prevalence and determinates of lung disease. Please clarify the prevalence of major lung diseases in this study, especially those that cause spirometric restriction.

The prevalence of non-communicable respiratory diseases (NCRDs) such as asthma, COPD and pulmonary fibrosis are not documented in Malawi due to limited diagnostic capacity in the country and a lack of studies conducted to determine the prevalence of major NCRDs. A few sentences to this effect have been added to page 5 of the manuscript along with supporting references.

Table 3 reports the proportion of participants at recruitment in 2014 to have respiratory symptoms, and those who had received a diagnosis of asthma or COPD previously.

2) For such a cohort study, it might make more sense to add the following: What is the diagnostic rate of COPD and asthma? How aware is the population of these diseases? How popular are lung function tests?

The proportion of our study participants who had been diagnosed with asthma and COPD at recruitment has been included in the manuscript in Table 3. 

The capacity to diagnose COPD and asthma is limited in Malawi because lung function testing in Malawi is not readily available out with research settings. In Chikwawa, lung function testing is not available at primary health facilities or secondary care (Chikwawa District Hospital). There is very limited capacity to perform spirometry in tertiary care (such as Queen Elizabeth Central Hospital), and this is provided by research staff, and for most patients, transport costs would prevent them from travelling to access this. On the other hand, the country has well-established TB control programme thus although the population is aware of respiratory symptoms knowledge and diagnosis of COPD and asthma may be limited. Additional text and citations have been included in the manuscript to explain this state. See page 5.

3) Analysis of lung function decline in different diseases and age stratification.

This paper has presented the overall rate of lung function decline of data collected in 2015 – 2017 (See page 18). In addition, we mention that there were no differences in the rate of decline between age groups or with diagnosed asthma or COPD (numbers were small).

Presently, we are have just finished collecting data for the most recent round of follow-up in August 2020 and are analysing that data. The data from a later time point to the baseline will provide better estimates for the annual rate of lung function decline and will be stratified by diagnosed lung disease and age when presented.

4) Analysis of reasons for reluctance to participate in follow-up.

This will be ascertained, and the results presented in subsequent publications. Analysis of a recent round of follow-up is going on at the moment.

5) There are obvious errors in the statistics of many demographic data, which need to be further verified in Table 3.

All the errors in Table 3 have been corrected. The statistical tests undertaken have also been explained so as to clarify our analysis (see page 8). We apologise if errors remain and would be willing to correct if requested by editorial team.

Reviewer #2: 

1) Purpose well stated but no justification- include justification (The WHY)

This has been added. See page 6.

2) Please include names of Ethical review members

The opinions of Ethics Committees are considered to be collective and we consider it unusual to list the names of three Review Committees (Exceeding 80 members in total). Instead we have provided links to the Imperial College Ethics Committee, Malawi College of Medicine ethical bodies and the Liverpool School of Tropical Medicine bodies that approved this study have been included in the manuscript. See page 16. 

3) What was the involvement of local researcher?

The study is presently housed within the Malawi-Liverpool Wellcome Trust Clinical Research Programme. MN, SR, RN, SG and JR are affiliated with MLW.

Additional MLW staff involved with the study have been acknowledged in the acknowledgement. See page 26

4) NHANES III – used no mention of limitation/ if appropriate or how it was adjusted for non-Caucasian population.

We have provided the estimates using the NHANES III African American reference ranges for the 2017 follow up. See Page 17. In the limitations we now mention the lack of widely accepted reference equations for sub-Saharan Africa and the noticeable difference in restriction when using the NHANES III equations for African-Americans when compared with American Caucasian equations. We also mention the use of GLI 2012 in the current round of follow-up.

5) With over 50% of participants decline to be recruited – there should be information of why? And state if this will bias the study.

Most of those initially identified to be approached but who did not participate were simply not traceable. We have presented the age and sex characteristics of those who were and were not recruited in 2014. We mention the possible biases associated with this in the limitations. One of the stipulations of the ethical approvals was/is that potential participants are able to refuse to take part without giving reason and therefore we are unable to comment on specific reasons why people declined to take part in the original 2014 study.

In the present round of follow-up, we have been able to follow-up >85% of the participants who participated in the baseline study. See page 18. The analysis of these data is still ongoing since we finished data collection for this round of follow-up in August 2020.

6) Please include results of lung function test done.

We have provided a minimal anonymised dataset containing the results of the lung function tests done. This has been included in the publication in the supplementary files.

Yours Sincerely,

Martin Njoroge (Corresponding Author)

E-mail: martin.njoroge@lstmed.ac.uk

---

## [Editor Report · Decision Letter 1]

29 Oct 2020

Cohort profile: The Chikwawa lung health cohort; a population-based observational non-communicable respiratory disease study of adults in Malawi.

PONE-D-20-12808R1

Dear Dr. Njoroge,

We’re pleased to inform you that your manuscript has been judged scientifically suitable for publication and will be formally accepted for publication once it meets all outstanding technical requirements.

Kind regards,

Alana T Brennan

Academic Editor

PLOS ONE
---

## [Editor Report · Acceptance letter]

4 Nov 2020

PONE-D-20-12808R1 

Cohort profile: The Chikwawa lung health cohort; a population-based observational non-communicable respiratory disease study of adults in Malawi. 

Dear Dr. Njoroge:

I'm pleased to inform you that your manuscript has been deemed suitable for publication in PLOS ONE. Congratulations! Your manuscript is now with our production department. 

Kind regards, 

on behalf of

Dr. Alana T Brennan 

Academic Editor

PLOS ONE